# Scaling Behaviors of LLM Reinforcement Learning Post-Training: An Empirical Study in Mathematical Reasoning

## Abstract

While scaling laws for large language models (LLMs) during pre-training have been extensively studied, their behavior under reinforcement learning (RL) post-training remains largely unexplored. This paper presents a systematic empirical investigation of scaling behaviors in RL-based post-training, with a particular focus on mathematical reasoning. Based on a set of experiments across the full Qwen2.5 dense model series (0.5B to 72B), we characterize how model scale, data volume, and computational budget interact to shape performance. Our analysis leads to four key findings: ❶ Under a fixed computational budget, larger models trained for fewer steps consistently outperform smaller models trained for more steps. ❷ Given a fixed amount of training data, larger models achieve superior sample efficiency, yielding lower loss. ❸ In data-constrained regimes, repeated reuse of high-quality data proves highly effective, as final performance is primarily governed by the total number of optimization steps rather than the uniqueness of samples. ❹ These scaling behaviors are robust across both base and instruction-tuned models, which share similar learning dynamics (e.g., larger models show faster convergence) even while differing in absolute accuracy. We further show that the relationship between test loss, compute, and data can be modeled by a predictive power-law with an analytic learning efficiency term $k(N)$ that demonstrates an efficiency saturation effect as model size increases. Collectively, these results provide a principled foundation and practical guidelines for efficiently scaling the reasoning capabilities of LLMs through RL post-training.

## 1 Introduction

The rapid progress of large language models (LLMs) has made elucidating their scaling laws a matter of central importance. These laws, which capture the intricate relationships between model architecture, parameter size, computational cost, data availability, and downstream performance (Kaplan et al., 2020; Hoffmann et al., 2022), are invaluable not only because they illuminate the latent factors governing learning dynamics, but also because they provide actionable guidance on how to distribute scarce computational resources most effectively (Li et al., 2025a). While extensive efforts have clarified scaling behavior, the scaling behavior of reinforcement learning (RL) post-training for LLM reasoning remains underexplored.

During pretraining, Kaplan et al. (2020) show that cross-entropy loss follows smooth power-law scaling in model size, dataset size, and training compute, implying that larger models trained for fewer steps are compute-optimal. Hoffmann et al. (2022) refine this by showing that, under fixed compute, scaling parameters and tokens proportionally is optimal, since many large models are undertrained. Extending to neural-based RL, Hilton et al. (2023) empirically demonstrates that the intrinsic performance of convolutional neural networks (CNNs) optimized via reinforcement learning also scales like power-law with model capacity and environment interaction.

Recently, RL has become the predominant post-training strategy for enhancing the reasoning abilities of LLMs, particularly in mathematics, a domain that demands long-horizon, compositional reasoning (Ferrag et al., 2025; DeepSeek-AI, 2025; Kimi Team, 2025; Ahn et al., 2024). Given the recent works applying RL for LLM reasoning, it is crucial to understand how to scale RL training

effectively for reasoning tasks. In this work, we aim to fill this research gap with an extensive empirical study on models of varying scales. Defining the normalized test error rate as Test Loss ($L$), we formalize the resource allocation challenge as a series of constrained optimization problems:

1. **The Compute-Constrained Scenario:** Given a fixed computational budget $C$, we seek the model size $N$ (and corresponding data allocation $D$) that minimizes the final loss:

$$\underset{N,\, D}{\arg\min}\; L(N, D) \quad \text{s.t.} \quad \text{FLOPs}(N, D) = C_{\text{const}}, \tag{1}$$

2. **The Data-Constrained Scenario:** For a fixed amount of unique training data $D$, we aim to determine the model size $N$ that yields the lowest test loss under this data limitation:

$$\underset{N,\, C}{\arg\min}\; L(N, C) \quad \text{s.t.} \quad D = D_{\text{const}}, \tag{2}$$

3. **The Data Reuse Scenario:** With a fixed number of optimization steps $S$, we investigate the impact of the data reuse factor $\tau$—the number of times each unique example is revisited—on the final loss. In this setting, the total number of processed samples is constrained as $D_{\text{unique}} \times \tau = S_{\text{const}}$. Formally:

$$\underset{\tau,\, C}{\arg\min}\; L(\tau, C) \quad \text{s.t.} \quad D_{\text{unique}} \times \tau = S_{\text{const}}, \tag{3}$$

To empirically answer these questions, we fine-tune 63 LLMs with reinforcement learning on 50k mathematics problems, based on the Qwen2.5 model family (Qwen et al., 2025). Figure 1 shows that, within the 0.5B-72B range, the loss reduction brought by RL follows an approximately log-linear trend with compute and data size. Importantly, larger models not only have better initial performance but also generally have more efficiency in computation and data utilization during the optimization process. Through deeper analysis, we find that the scaling behavior exhibits meaningful predictivity, and further reveals that the efficiency gains from increasing model scale diminish gradually, leading to a saturation effect for larger-scale models.

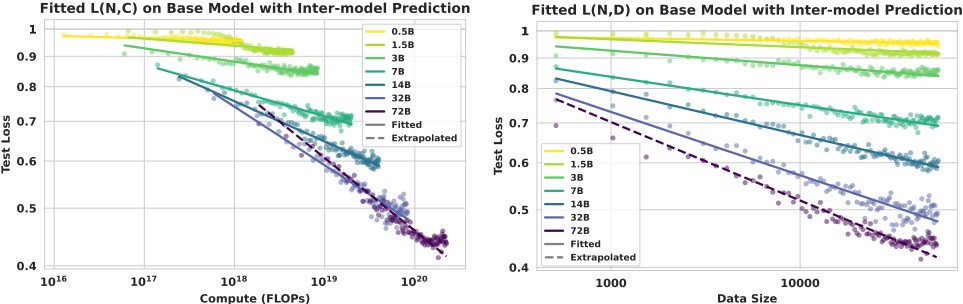

(a) Relation between test loss and training compute for for base models, fitted on 0.5B-32B and extrapolated on 72B.

(b) Relation between test loss and data size for base model, fitted on 0.5B-32B and extrapolated on 72B.

Figure 1: Training Data Fiited on 0.5B-32B with Extrapolation on 72B: In both cases, larger models consistently exhibit higher learning efficiency than smaller models.

We further analyze the data-constrained regime, where we demonstrate that data reuse is a highly effective strategy. We validate the generality of our findings through extensive ablation studies on both base and instruct model series. Besides, we also study the impact of the rollout number in the GRPO algorithm (Shao et al., 2024). These investigations establish fundamental scaling relationships for RL post-training, providing a quantitative foundation and practical guidelines for resource-efficient model refinement.

Specifically, our key findings can be summarized as follows:

- In our experiment scale, larger models, starting with stronger initial performance, consistently achieve better compute and data efficiency in RL post-training for mathematical reasoning. However, the marginal gains in this efficiency diminish gradually, revealing a saturation trend as model scale increases.

- The scaling law exhibits predictive capability, allowing us to forecast the training efficiency of larger models and predict the remaining training trajectory from early training data.
- In data-limited settings, repeated exposure to a small dataset is nearly as effective as using larger corpora, highlighting data reuse as a practical strategy.
- In math reasoning, scaling behaviors are robust across both base and instruction models and GRPO hyperparameters, providing actionable guidance for resource-efficient RL post-training.

## 2 EXPERIMENTAL SETUP

We describe the experimental setup for studying scaling behavior in RL post-training of LLMs for mathematical reasoning, including the model family, training and evaluation data, and evaluation protocol in this section. Full details are provided in Appendix A.

**Models and Framework.** We use the Qwen2.5 model family (0.5B, 1.5B, 3B, 7B, and 14B parameters) (Qwen et al., 2025), which shares the same architecture, so that parameter count is the only variable in our scaling analysis. All experiments are run with the VeRL framework (Sheng et al., 2024), a large-scale RL platform for LLMs ensuring consistency and reproducibility.

**RL Algorithm.** We use Group Relative Policy Optimization (GRPO) (Shao et al., 2024) for RL fine-tuning. GRPO estimates advantages by normalizing rewards across responses sampled from the same prompt, yielding a stable signal with lower memory cost. Specifically, for each question $q$, GRPO samples a group of outputs $\{o_1, o_2, \cdots, o_G\}$ from the old policy $\pi_{\theta_{old}}$, and the objective is defined as

$$\mathcal{L}_{\text{GRPO}} = \frac{1}{G} \sum_{i=1}^{G} \frac{1}{|o_i|} \sum_{t=1}^{|o_i|} \Big\{ \min\Big[ \rho(\theta)\, \hat{A}_{i,t},\, \text{clip}\Big( \rho(\theta), 1 - \varepsilon,\, 1 + \varepsilon \Big) \hat{A}_{i,t} \Big] - \beta\, \text{D}_{\text{KL}} \Big\}, \quad (4)$$

w here $\rho(\theta) = \frac{\pi_\theta(o_{i,t}|q, o_{i,<t})}{\pi_{\theta_{\text{old}}}(o_{i,t}|q, o_{i,<t})}$ is the important sampling weight. For each output $o_i$, a reward model or rule is used to yield the reward signal $\mathbf{r} = \{r_1, r_2, \cdots, r_G\}$. The advantage is computed as

$$\hat{A}_{i,t} = \frac{r_i - \text{mean}(\mathbf{r})}{\text{std}(\mathbf{r})}. \quad (5)$$

**Dataset settings.** The training data is the mathematics subset of the `guru-RL-92k` dataset from the Reasoning360 project (Cheng et al., 2025), which is carefully curated through deduplication and difficulty filtering. We further sort the problems by increasing difficulty (decreasing pass rate, evaluated by Qwen2.5-7B model) to enable curriculum learning. The evaluation data consists of two parts. To derive scaling laws, we use a held-out set of 500 in-domain math problems sampled from the training distribution. To assess generalization, we evaluate on a broader benchmark suite spanning mathematics (AIME2024 (Patel et al., 2024), AMC2023 (KnovelEng, 2025), GSM8K (Cobbe et al., 2021), MATH500 (Lightman et al., 2023)), code (HumanEval (Chen et al., 2021)), logic (Zebra Puzzle(Lin, 2024)), and science (SuperGPQA (Team et al., 2025)). More details about dataset settings can be found in Appendix A.1.

**Prompt Setting.** To ensure stable behavior during RL training and evaluation, we use structured prompts tailored to each domain. For example, all mathematics problems are prepended with the Chain-of-Thought prompt (Wei et al., 2023): *"You are a knowledgeable math assistant. Answer the following questions and think step by step"*. More prompt templates for all related domains could be found in Appendix A.3.

**Evaluation Process.** We compute the Pass@1 score using a binary reward signal derived from a deterministic, rule-based process. For each problem, a script extracts the final answer from the model output (e.g., within a \boxed{} for math) and compares it to the ground truth. A reward of 1 is given for a correct match and 0 otherwise. This signal is not only used to calculate test loss during evaluation, but also as the reward during RL training.

**Metric.** Our primary evaluation metric is the test loss ($L$), a proxy for reward-based performance in the RL setting. Formally, $L = 1 - (R/R_{max})$, where $R$ is the number of correct solutions and $R_{max}$ the total. We adopt the term "test loss" for consistency with foundational neural scaling law literature (Kaplan et al. (2020)). Obviously, the goal of maximizing reward in RL training is equivalent to minimizing the test loss $L$.

# 3 EMPIRICAL RESULTS AND SCALING LAWS

This section presents a comprehensive empirical investigation into the scaling behavior of RL for post-training LLMs. Our experiments are designed to address the core questions posed in Eq. 1, Eq. 2, and Eq. 3. We first examine scaling behaviors under compute and data constraints, then analyze independent scaling dimensions, data reuse strategies, and finally evaluate generalization performance together with an ablation study on the GRPO group size $G$ (Eq. 4). To ensure robust conclusions, each configuration is repeated **three times** for both base and instruct models. Their statistical uncertainty analysis, including Average Standard Deviation and Standard Error of the Mean (SEM), are provided in Appendix C.3.

## 3.1 COMPUTE-OPTIMAL SCALING

> **Observation 1**
>
> Within the model parameter range of 0.5B to 72B, RL post-training under a fixed computational budget $C$ is compute-optimal when prioritizing the training of larger models rather than extending the training duration of smaller models.

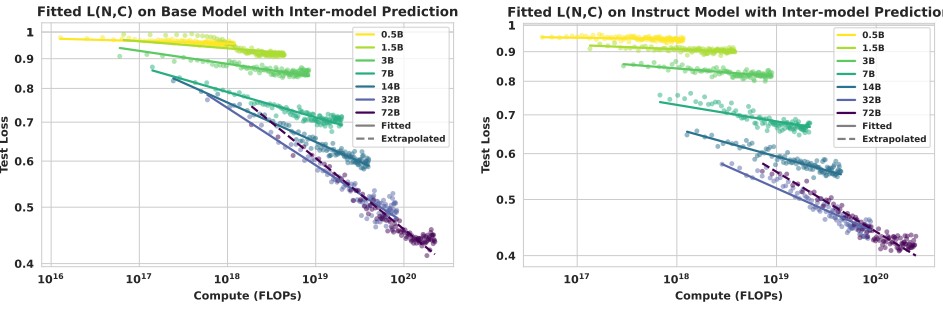

(a) Test loss vs training compute with extrapolation on 72B for base model
(b) Test loss vs training compute with extrapolation on 72B for instruct model

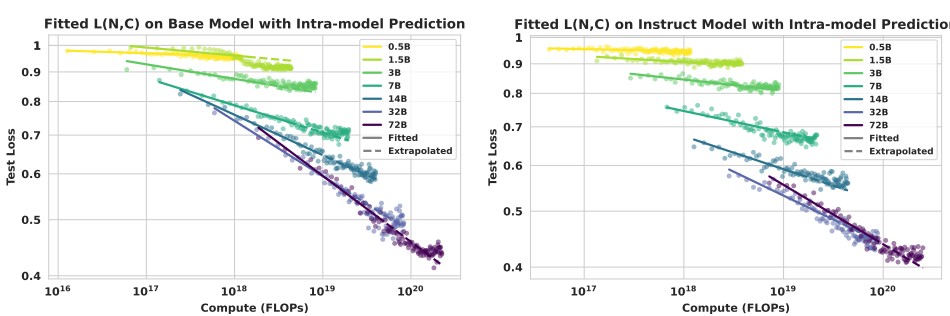

(c) Test loss vs training compute with extrapolation for remainder of each training process on base model
(d) Test loss vs training compute with extrapolation for remainder of each training process on instruct model

Figure 2: Compute Scaling and Predictive Capability from 0.5B-72B for Base and Instruct Models

In the compute-constrained setting (Eq. 1), we train 0.5B–14B models and measure test loss as a function of cumulative FLOPs $C$. As shown in Figure 2, larger models consistently outperform

smaller ones under the same compute budget for both base and instruct variants. These plots include both Inter-model Extrapolation (fitted on 0.5B-32B and extrapolated on 72B) and Intra-model Prediction (predicting the remainder of training from initial steps) to demonstrate the predictive power of our derived scaling law. The loss–compute relationship follows a log-linear trend, which can be modeled by a power law:

$$\log(L(N,C)) = -k_C(N) \cdot \log(C) + E_C(N), \text{ where } k_C(N) = \left( \frac{K_{Cmax}}{1 + \frac{N_C}{N}} \right) \quad (6)$$

To demonstrate the predictive capability of the proposed formula Eq 6, we evaluate it in two distinct extrapolation settings:

1. Inter-model Extrapolation: We fit the law's parameters on smaller models (0.5B–32B) to calculate the learning efficiency ($k_C(N)$) of 72B model. As shown in Figure 2a and 2b, the predicted efficiency aligns closely with the actual 72B performance.

2. Intra-model Prediction: We fit the law using only early training steps to forecast the remaining trajectory for a specific model, shown in Figure 2c and 2d.

We further analyze learning efficiency term $k_C(N)$ in Eq. 6. As Figure 4a shows, $k_C(N)$ grows with model size $N$, meaning larger models consistently have higher learning efficiency. However, the efficiency gain from model scale is not uniformly linear. Beyond 32B, the increase in $k_C(N)$ diminishes, leading to efficiency saturation.

## 3.2 DATA-OPTIMAL SCALING

> **Observation 2**
>
> Within the model range of 0.5B to 72B, for a fixed volume of unique training data $D$, larger models demonstrate superior sample efficiency, consistently achieving lower test loss.

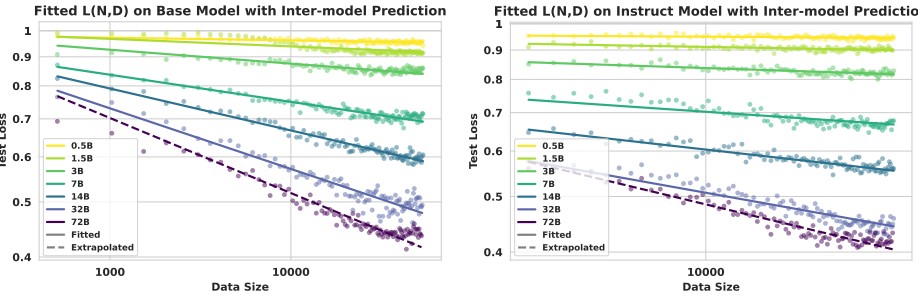

(a) Test loss vs data size with extrapolation on 72B for base model  (b) Test loss vs data size with extrapolation on 72B for instruct model

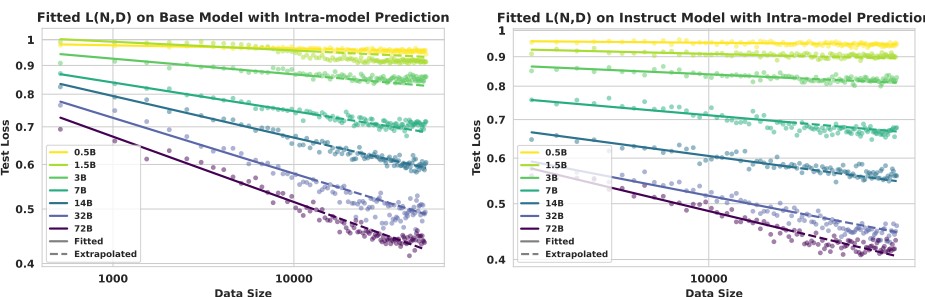

(c) Test loss vs data size with extrapolation for remainder of each training process on base model  (d) Test loss vs data size with extrapolation for remainder of each training process on instruct model

Figure 3: Data Scaling and Predictive Capability from 0.5B-72B for Base and Instruct Models

In the data-constrained setting (Eq. 2), we train models with varying parameter counts $N$ on fixed amounts of unique samples $D$. As shown in Figure 3, larger models consistently achieve lower test loss and higher sample efficiency across both base and instruct variants. The loss–data relationship follows a power law similar to the compute setting:

$$\log(L(N, D)) = -k_D(N) \cdot \log(D) + E_D(N), \text{ where } k_D(N) = \left( \frac{K_{Dmax}}{1 + \frac{N_D}{N}} \right) \quad (7)$$

Mirroring the analysis in Section 3.1, we evaluate the extrapolative capability of our data scaling law (Eq. 7) in two settings:

1. Inter-model Extrapolation: By fitting parameters on smaller models (0.5B–32B), we accurately predict the data efficiency ($k_D(N)$) on 72B model, as illustrated in Figure 3a and 3b.

2. Intra-model Prediction: We forecast the loss trajectory for the remainder of the training process using only early-stage data, shown in Figure 3c and 3d.

We adopt the same analytic form for the data efficiency coefficient $k_D(N)$ as we did for compute. As illustrated in Figure 4b, $k_D(N)$ follows a saturation curve identical to $k_C(N)$: while larger models excel at extracting knowledge from each data point, the efficiency gains diminish at scales beyond 32B. The unified functional form across both compute and data domains reflects the theoretical consistency of our scaling law.

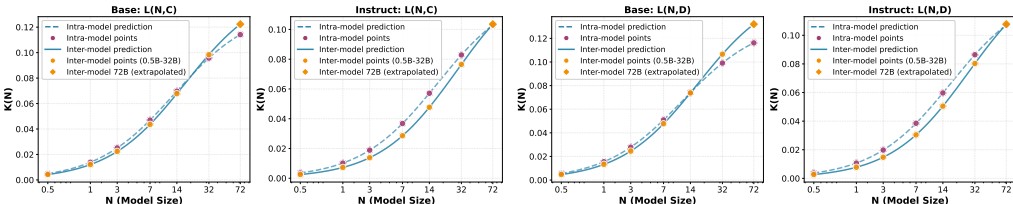

(a) Fitted learning efficiency coefficient $k(N)$ for L(N,C)

(b) Fitted learning efficiency coefficient $k(N)$ for L(N,D)

Figure 4: Fitted learning efficiency coefficient $k(N)$ for Base and Instruct models: Intra-model and inter-model predictions exhibit nearly identical growth trends, with efficiency gains starting to diminish beyond the 32B model.

## 3.3 SCALING UP MODEL SIZE

> **Observation 3**
>
> When trained to converge on sufficiently large datasets, test loss decreases monotonically with model size, though the trend deviates from a strict power law.

We train models of varying sizes to convergence and compare their final test loss. As shown in Figure 5a, larger models consistently achieve lower loss, improving monotonically with scale. The curve deviates from a strict power law: smaller models show weaker gains, suggesting diminishing returns at low parameter counts. A likely reason is that larger models inherit richer pre-trained representations, which reinforcement fine-tuning exploits for greater improvements than parameter growth alone would predict. Figure 5b further shows that as RL training progresses, larger models generate longer responses except for 32B model. This correlates with higher accuracy, indicating greater *test-time scaling efficiency*: additional inference tokens yield larger gains in bigger models.

We also benchmark our RL-tuned Qwen2.5 models (Qwen et al., 2025) against state-of-the-art open-source reasoning systems, including Qwen3 (Yang et al., 2025) and GPT-OSS (OpenAI et al., 2025), detailed in Table 4. On our held-out set, the 32B and 72B models match or surpass dense Qwen3 counterparts of similar size, highlighting the effectiveness of RL post-training. Mixture-of-experts models such as Qwen3 and GPT-OSS achieve approximate loss at much larger scales (235B), with

GPT-OSS-120B currently leading. These comparisons suggest that scaling across 0.5B-72B will be necessary to fully characterize post-training behavior and compete with frontier MoE systems.

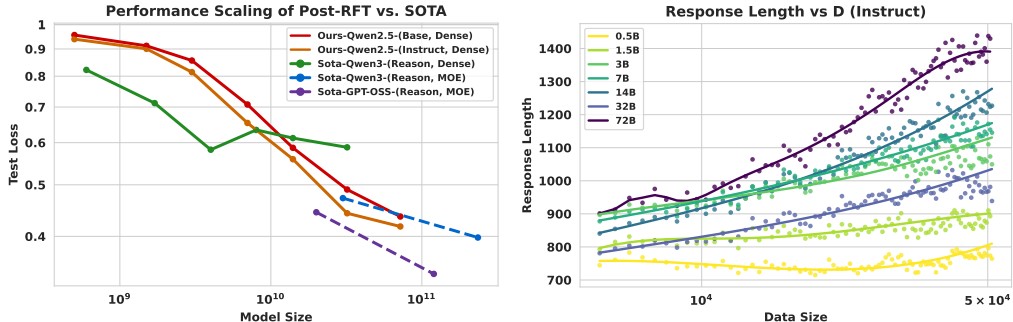

(a) Fig (a) shows Relation between test loss and model size $N$ for our trained model and SOTA models demonstrates the effectiveness of our training.

(b) Fig (b) shows relation between response length and data size for instruct models,larger models generate longer responses except for 32B model.

Figure 5: Analysis of model scaling properties shows the effectiveness of our training process in this empirical study.

### 3.4 SCALING WITH CONSTRAINED DATA AND REUSE

> **Observation 4**
>
> Performance in data-constrained settings is primarily determined by the total number of parameter update steps $(S)$. For a fixed $S$, the final test loss is remarkably insensitive to the data reuse factor $(\tau)$, with no significant degradation up to $\tau = 25$.

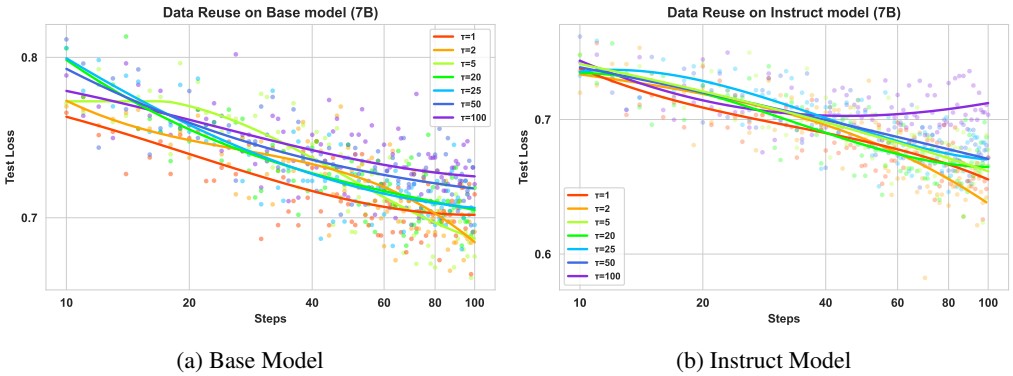

(a) Base Model

(b) Instruct Model

Figure 6: Plot shows the final test loss of a 7B Base (left) and Instruct (right) model trained with a fixed toal number of samples but varying the data reuse factor $\tau$.

We further consider the data-reuse scenario (Eq. 3), where high-quality data is limited but can be revisited multiple times. To simulate this, we partition the training set into smaller subsets while preserving the difficulty distribution (Details provided in Appendix A.4). Each subset is cycled through multiple times, with the reuse factor $\tau$ controlling how often each unique example is revisited. The total number of update steps $S$ is fixed across runs, and curriculum ordering is maintained so that problems are always presented from easy to hard. This ensures that performance differences arise solely from the degree of data reuse, rather than distributional or scheduling artifacts.

As shown in Figure 6, performance remains nearly unchanged for $\tau \leq 25$, while moderate degradation appears as $\tau$ increases further. At $\tau = 100$, we observe clear signs of overfitting, indicating that repeated reuse eventually harms generalization. Overall, these results suggest that final performance is primarily governed by the total number of optimization steps rather than sample uniqueness, and that moderate data reuse is an effective strategy for RL fine-tuning with limited datasets.

## 3.5 DOMAIN TRANSFER

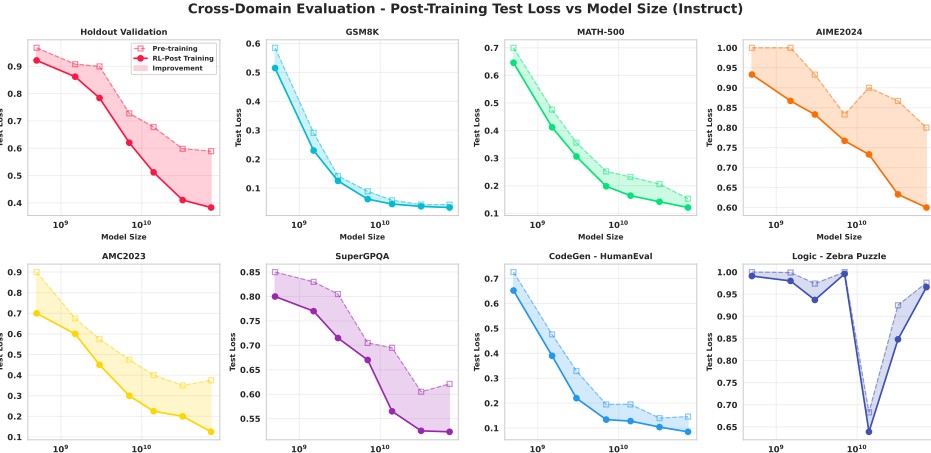

Figure 7: The effect of domain transfer, illustrated with the Qwen2.5-72B-Instruct model.

> **Observation 5**
>
> RL post-training on mathematical reasoning yields generalization improvements on in-domain tasks with varying difficulty, but shows negligible transfer to out-of-domain tasks.

We also investigate the generalization capabilities of the reinforcement learning fine-tuning (RFT) models by evaluating them on a suite of unseen in-domain tasks with varying difficulty and out-of-domain (OOD) tasks. More results are in Appendix B.1.

**In-Domain Generalization.** Figure 7 shows consistent improvements on unseen mathematics tasks outside the training set. On benchmarks, from easy to hard, including `GSM8K`, `MATH-500`, `AMC2023`, `AIME2024`, test loss steadily decreases with training compute, suggesting that RL post-training enhances transferable reasoning skills within mathematics.

**Out-of-Domain Generalization.** As shown in Figure 7, results on OOD tasks are markedly different. For code generation (`HumanEval`) and STEM problems (`SuperGPQA`), performance gains marginally, indicating that RL fine-tuning is highly specialized. On logical reasoning (`zebra_puzzle`), performance degrades for larger models, suggesting that intensive optimization on mathematical reasoning may interfere with or "damage" other distinct reasoning abilities.

## 3.6 ABLATION ON GRPO HYPERPARAMETERS

> **Observation 6**
>
> A larger GRPO rollout group size ($G$) is consistently more data-efficient, while the compute-optimal $G$ grows with the total computational budget.

We conducted an ablation study on the rollout group size $G$, a key GRPO hyperparameter that controls how many responses are sampled per prompt. This directly affects both the compute per update and the stability of the training signal. We tested $G \in \{4, 8, 16, 32\}$ on the 7B models.

**Data-centric View**. Figure 8b and 8d shows that larger rollout sizes consistently yield better sample efficiency: $G = 32$ achieves the lowest test loss for the same number of unique samples. This supports the intuition that more responses per question provide a stronger advantage estimate and thus more effective gradient updates.

**Compute-centric View**. The optimal rollout size $G$ is not fixed but shifts with the training budget. This implies that practitioners should tune $G$ according to available compute rather than relying on a universal setting. We attribute this dynamic to the trade-off between the higher variance reduction from larger $G$ and the additional FLOPs it consumes, which makes small $G$ preferable at low budgets but large $G$ superior when ample compute is available.

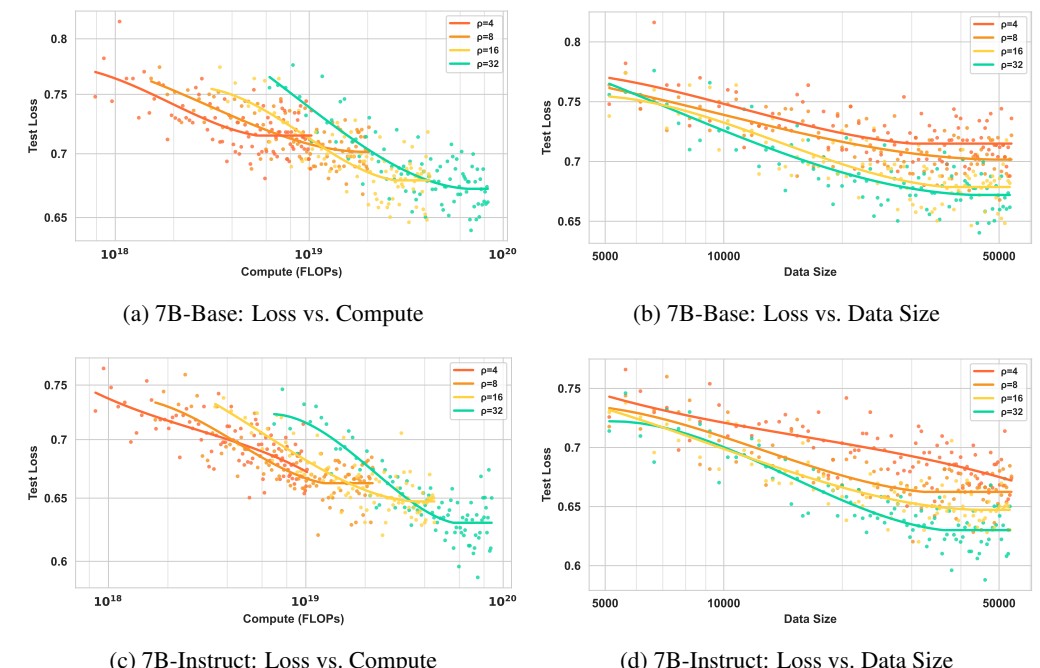

Figure 8: Effects of GRPO rollout size on training efficiency: (a–b) show compute/data scaling for 7B-Base; (c–d) for 7B-Instruct.

## 4 RELATED WORK

**Foundational Scaling Laws of Neural Language Models.** Foundational scaling studies show language modeling loss follows smooth power-laws in model size $N$, data $D$, and compute $C$ (Kaplan et al., 2020), with compute-optimal training prescribing near lockstep growth of parameters and tokens under fixed FLOPs (Hoffmann et al., 2022). Later analyses attribute earlier discrepancies to embedding/non-embedding parameter accounting, last-layer costs, optimizer warmup, and scale-sensitive hyperparameters (Pearce & Song, 2024; Porian et al., 2024), while data-centric refinements examine pruning efficiency (Sorscher et al., 2022), repetition effects (Hernandez et al., 2022), gzip-based complexity predictors (Pandey, 2024), constrained or synthetic regimes (Muennighoff et al., 2023; Qin et al., 2025), and task transfer (e.g., translation) (Isik et al., 2024). Test-time compute amplification supplies an inference analogue to classical training laws (Snell et al., 2024).

**RL post-training in LLMs.** In RL, power-law trends similarly link capacity, interaction compute, and performance (Hilton et al., 2023); scaling RFT across horizon and compute improves mathematical and coding reasoning (DeepSeek-AI, 2025; Kimi Team, 2025; Mai et al., 2025b; Zhang et al., 2025a;b), while extended schedules (Liu et al., 2025), ultra-low-shot or single-example RL (Wang et al., 2025), and minimal-data efficiency paradigms (Li et al., 2025b) probe data–compute tradeoffs. Instability and uneven gains highlight fragile optimization (Zeng et al., 2025a; Yue et al., 2025), and multi-domain mixtures reveal both synergy and interference across math, code, and logic (Li et al., 2025c; Cheng et al., 2025).

**Mathematical Reasoning with LLMs.** Mathematical reasoning amplifies these dynamics: accuracy generally scales upward while verification behaviors remain inconsistent (Touvron et al., 2023); corpus volume and quality jointly shape attainable curves (Ye et al., 2024); multi-task math-generalist training diverges from specialist scaling trajectories (Yue et al., 2023); and RL with code execution induces additional behaviors such as emergent tool use concentrated in math problem solving (Zeng et al., 2025b). Collectively, evidence indicates that reasoning performance is governed by interacting axes of model size, data distribution/quality, training (supervised vs. RL) paradigm, and allocation of both training and inference compute, while unified laws for mathematical reasoning remain only partially characterized.

## 5 DISCUSSION

**Scaling Dependence on Evaluation Environment and Metrics.** Reinforcement learning optimizes directly for environment rewards (Sutton & Barto, 2018), which in principle allows unbounded capability—as demonstrated by AlphaZero mastering board games (Silver et al., 2017), AlphaFold predicting protein structures (Jumper et al., 2021), and frontier LLMs such as Gemini-2.5-Pro achieving IMO-level performance (Huang & Yang, 2025). In contrast, text-based LLMs lack a well-defined RL environment, forcing us to rely on human-curated datasets as proxies. Test loss thus serves as a pragmatic but imperfect metric: it is monotonic and convergent, yet heavily dependent on dataset construction and task difficulty, with different benchmarks (e.g., GSM8K vs. AIME, Section 3.5) showing distinct convergence rates. This task dependence makes the absolute coefficients of our fitted scaling laws ($k(N), E$) difficult to interpret universally. Prior work proposed "intrinsic performance"—the minimum compute needed to reach a target reward—as a normalization across environments (Hilton et al., 2023), but we did not find an analogous measure in large-scale LLMs. Establishing principled, environment-independent evaluation protocols remains an open and critical challenge for RL-based scaling studies.

**Scaling Dependence on Model Scale.** Our study of models from 0.5B to 72B parameters shows that larger models exhibit greater sample and compute efficiency in RL post-training. This parameter range allows us to precisely characterize the scaling limits. We found that these advantages do not extend indefinitely. Our analytic learning efficiency term $k(N)$ in Eq.6 and Eq.7, explicitly confirms that the efficiency gains follow a saturation curve toward a limit ($K_{\max}$). This finding implies that scaling up models beyond a certain point, while still yielding absolute performance gains, suffers from diminishing marginal returns in efficiency.

**Dependence on RL Algorithm.** Our analysis is based on GRPO, a mainstream and stable RL post-training algorithm for LLMs that uses an actor-only design and normalizes rewards across responses. Comparative study with alternative RL algorithms (Cui et al., 2025) reports minor differences in training curves. Whether more advanced algorithms can significantly improve sample efficiency or stability—and thereby reshape the scaling frontier—remains an important open question.

**Future of LLM Agent.** The integration of reinforcement learning with agentic LLMs is increasingly viewed as a promising direction (Zhang et al., 2025a;b). Both theoretical and empirical studies show that augmentations such as external tool use and long-term memory can substantially boost model performance (Lin & Xu, 2025; Houliston et al., 2025; Mai et al., 2025a). We anticipate that such agentic mechanisms will markedly improve the scaling behavior of RL-trained LLMs: by offloading deterministic computations to tools and focusing learning on high-level decision making, these models could achieve much higher efficiency, effectively shifting the performance frontier upward for a given compute or data budget. Understanding the scaling laws of these agentic systems is, therefore, a key and exciting avenue for future research.

## 6 CONCLUSION

This study presents the systematic exploration of scaling laws for reinforcement learning post-training of large language models in mathematical reasoning. Through 54 controlled experiments across the Qwen2.5 family, we show that larger models consistently achieve superior compute and data efficiency, that performance in data-limited regimes depends primarily on the total number of optimization steps rather than data uniqueness, and that moderate data reuse can be highly effective without harming generalization. While RL post-training reliably strengthens in-domain reasoning, its transfer to out-of-domain tasks remains limited, underscoring the trade-off between specialization and breadth. Our ablation further identifies rollout size in GRPO as a practical lever tied to compute budgets. Taken together, these findings offer principled and actionable guidelines for resource-efficient RL fine-tuning and suggest promising directions for further exploration of scaling and generalization in LLM reasoning.

# 7 STATEMENT

## 7.1 ETHICS STATEMENT

The authors have read and adhered to the ICLR Code of Ethics. This work is foundational in nature, focusing on the scaling properties of large language models in the domain of mathematical reasoning. Our research exclusively utilizes publicly available and previously published resources, including open-source models (e.g., Qwen2.5) and established datasets (e.g.,guru-RL-92k), thereby mitigating concerns related to data privacy, human subjects, or the release of sensitive information. The application domain of mathematical problem-solving does not inherently present risks of direct societal harm. The primary ethical consideration associated with this work is the environmental impact of the computational resources required for large-scale model training, a challenge common to the field. We believe that by providing insights into efficient resource allocation, our work contributes positively to mitigating this concern for future research.

## 7.2 REPRODUCIBILITY STATEMENT

We are committed to ensuring the reproducibility of our research. All aspects of our experimental design are detailed to facilitate this. The specific models used (Qwen2.5 series), are described in Section 2. The training and evaluation datasets, including our held-out set, are also detailed in Section 2, with a full breakdown provided in Appendix A.1. Our reinforcement learning methodology is based on the GRPO algorithm, and all relevant hyperparameters used for training and evaluation are listed in Appendix A.2. The evaluation protocol, including the precise definition of our Test Loss metric and the prompt templates for all tasks, is outlined in Section 2 and Appendix A.3. All figures and observations presented are directly generated from the raw data collected during our experiments. At last, we provide the experiment source code in Supplementary Material along with the submission.

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

# A EXPERIMENT SETUP DETAILS

This section provides a detailed breakdown of the datasets and hyperparameters used in our experiments, supplementing the information provided in the main text.

## A.1 DATASET DETAILS

Our training was conducted on a curated mathematics dataset. For evaluation, especially for analyzing generalization (as mentioned in the main text), we utilized a comprehensive suite of benchmarks spanning multiple domains. The composition of this evaluation suite is detailed in Table 1.

Table 1: Composition of the multi-domain evaluation suite.

| Dataset | Samples | Huggingface Tag | Domain |
|---|---|---|---|
| Held-out Data | 500 | LLM360/guru-RL-92k | Math |
| aime2024 | 30 | Maxwell-Jia/AIME_2024 | Math |
| amc2023 | 40 | knoveleng/AMC-23 | Math |
| codegen_humaneval | 164 | openai/openai_humaneval | Code |
| gsm8k | 1319 | openai/gsm8k | Math |
| logic_zebra_puzzle | 200 | LLM360/guru-RL-92k | Logical Reasoning |
| math | 500 | HuggingFaceH4/MATH-500 | Math |
| stem_supergpqa | 200 | LLM360/guru-RL-92k | STEM |
| **Total** | **2953** | | |

## A.2 HYPERPARAMETER CONFIGURATION

All experiments were conducted with a consistent set of hyperparameters for the Group Relative Policy Optimization (GRPO) algorithm to ensure a fair comparison across different model sizes and configurations. The key hyperparameters are listed in Table 2.

Table 2: GRPO training hyperparameters used across all experiments.

| Hyperparameter | Value |
|---|---|
| Learning Rate | $1.0 \times 10^{-6}$ |
| Batch Size | 512 |
| KL Loss Coefficient | 0.001 |
| Rollout Temperature (Training) | 1.0 |
| Rollout Temperature (Evaluation) | 0.7 |
| Clip Ratio (High & Low) | 0.2 |
| Input Sequence Length | 2048 |
| Output Sequence Length | 4096 |

## A.3 PROMPT TEMPLATES

This section details the specific prompt templates used for evaluating models on different domains. For each task, the model was provided with the corresponding instruction prepended to the problem statement `<question>`.

Table 3: Prompt templates used for different evaluation domains.

| Domain | Prompt Template |
|---|---|
| Mathematics | You are a knowledgeable math assistant. Answer the following questions and think step by step\n<question>\nPlease output the final answer within \\boxed{}. |
| Code | Write a complete, self-contained Python solution to the following problem. Your solution must include all necessary imports and the full function definition, including the signature exactly as specified. Do not modify the function signature or docstring.\n<question> |
| Logic | Solve the following puzzle\n<question>\nPlease return the final answer in <answer> </answer> tags, for example <answer> {"header": ["Position", "Nationality", "Job"], "rows": [["1", "british", "plumber"], ["2", "polish", "carpenter"]]} </answer>. |
| Science (STEM) | You are a knowledgeable assistant. Answer the following questions and think step by step \n <question> \n put your final answer option within \\boxed{}. Only put the letter in the box, e.g. \\boxed{A}. There is only one correct answer |

## A.4 DATA REUSE EXPERIMENT SETUP

To systematically evaluate the effect of data reuse under constrained data scenarios, we design controlled experiments where all runs are trained with the same total data size but different levels of data repetition. Each run randomly samples a subset from the full training corpus and repeats this subset sufficiently many times to exactly match the target data budget (i.e., subset size $\times \tau$ = total data size). Unlike Muennighoff et al. (2023), subsets are sampled independently for each run rather than sampling within the larger subsets, to mitigate sampling bias and balance stochasticity across conditions. To remain consistent with the Curriculum Learning setting of the main experiments, examples within each subset are ordered by increasing difficulty; across epochs, this difficulty schedule is preserved and repeated rather than reshuffled, as illustrated in Figure 9.

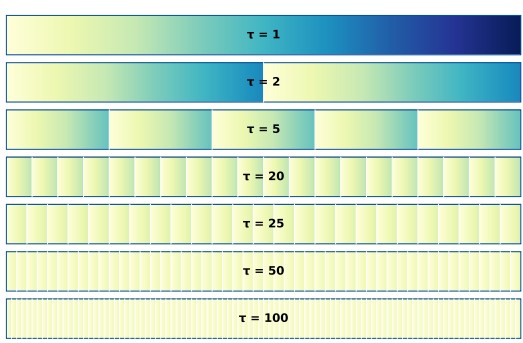

Figure 9: Data Reuse Schema

# B ADDITIONAL EXPERIMENT RESULTS

This section provides supplementary experimental results that support and extend the analyses presented in the main body of the paper.

## B.1 PERFORMANCE ON IN-DOMAIN AND OUT-OF-DOMAIN TASKS

To assess how the mathematical reasoning capabilities acquired during RL fine-tuning generalize, we evaluated our models on a comprehensive suite of unseen benchmarks. We categorize these into two groups: in-domain different tasks (other mathematics datasets) and out-of-domain tasks (e.g., code, science, logic). The results are presented in Figure 10 and Figure 11.

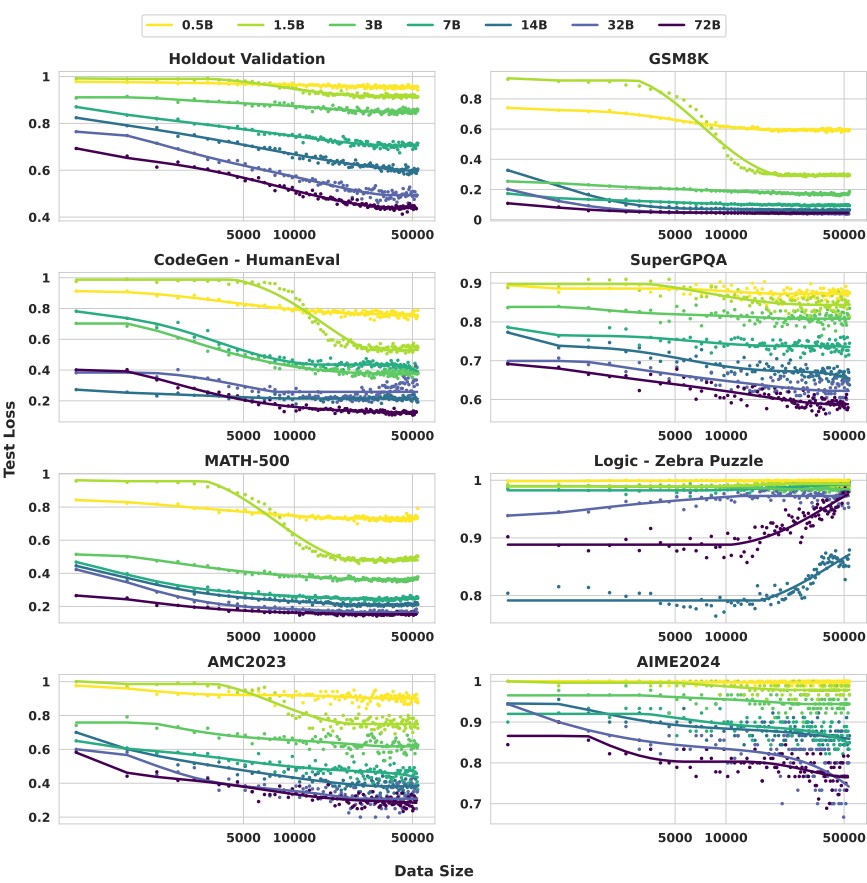

Figure 10: Test loss on in-domain and out-of-domain benchmarks vs data size for Base models. It shows modest positive transfer on in-domain tasks, with limited or negative transfer on OOD tasks.

**In-Domain Generalization (Different Mathematical Tasks).** On mathematics benchmarks not included in our training set (such as GSM8K, MATH, AIME, and AMC), we observe a generally positive transfer of learned skills. For most of these tasks, the test loss shows a modest but consistent decrease as training progresses, particularly for the larger models. This suggests that the model's enhanced reasoning ability is not overfitted to the training distribution and is applicable to a wider range of mathematical problems.

**Out-of-Domain Generalization.** When evaluating on tasks outside of mathematics, the generalization is more limited. For both code generation (HumanEval) and science problems (SuperGPQA), performance remains largely static throughout the training process across all model sizes, with test loss curves staying flat. This indicates that the specialized mathematical reasoning skills do not readily transfer to these domains. A noteworthy phenomenon is observed in the logical reasoning task (Zebra Puzzle): the largest models (particularly the 14B variants) show a degradation in performance (an increase in test loss) as training progresses, suggesting a potential negative transfer effect where intensive optimization on mathematical reasoning may interfere with capabilities required for certain types of logical puzzles.

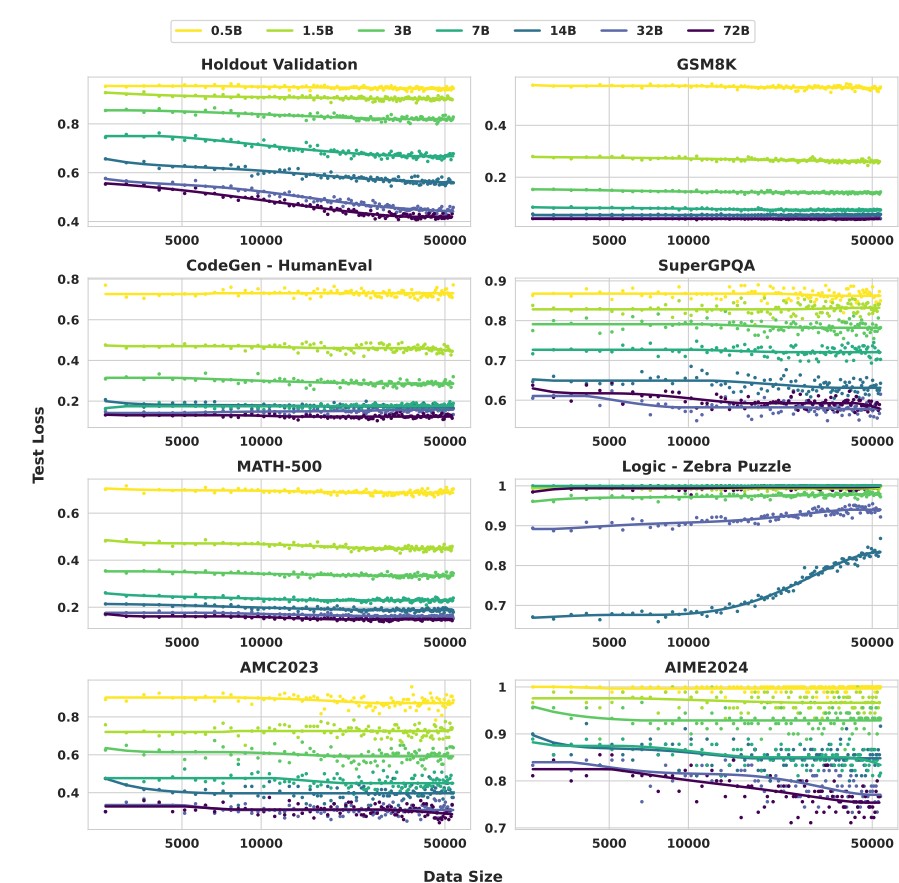

Figure 11: Test loss on in-domain and out-of-domain benchmarks vs data size for Instruct models. It shows modest positive transfer on in-domain tasks, with limited or negative transfer on OOD tasks.

### B.2 PEFORMANCE COMPARED WITH ADVANCE MODELS

| Model Family | Model Identifier | Pass@1 Score |
|---|---|---|
| *Models from Our Study (Post-RFT)* | | |
| Qwen2.5-Base | 0.5B | 0.070 |
| | 1.5B | 0.116 |
| | 3B | 0.182 |
| | 7B | 0.338 |
| | 14B | 0.450 |
| | 32B | 0.540 |
| | 72B | 0.607 |
| Qwen2.5-Instruct | 0.5B | 0.078 |
| | 1.5B | 0.138 |
| | 3B | 0.216 |
| | 7B | 0.380 |
| | 14B | 0.488 |
| | 32B | 0.590 |
| | 72B | 0.617 |
| *External SOTA Models (for Comparison)* | | |
| Qwen3 | 0.6B | 0.178 |
| | 1.7B | 0.288 |
| | 4B | 0.418 |
| | 8B | 0.366 |
| | 14B | 0.388 |
| | 30B (A3B) | 0.528 |
| | 32B | 0.412 |
| | 235B (A22B) | **0.602** |
| GPT-OSS | 20B | 0.556 |
| | 120B | **0.660** |

Table 4: Performance of various models on the held-out evaluation set.

To contextualize the performance of our models and the difficulty of our primary evaluation metric, we benchmarked a range of external, state-of-the-art (SOTA) models on our held-out mathematics test set. The results are presented in Table 4. The performance of our Qwen2.5 models reflects their final scores after the completion of reinforcement learning fine-tuning (RFT), while others are benchmarked directly.

## C FORMULA FITTING AND DERIVATION

### C.1 FLOPS CALCULATION METHODOLOGY

The computational cost for a LLM is primarily determined by the number of non-embedding parameters ($N$) and the number of processed tokens ($T$). The costs for the fundamental operations are:

- **Forward Pass Cost:** The cost of a single forward pass is approximately $C_{\text{fwd}} \approx 2NT$ FLOPs.

- **Backward Pass Cost:** The backward pass is approximately twice as expensive as the forward pass, so $C_{\text{bwd}} \approx 4NT$ FLOPs.

A full training step, which includes one forward and one backward pass for the gradient update, therefore has a total computational cost of:

$$C_{\text{train}} = C_{\text{fwd}} + C_{\text{bwd}} \approx 2NT + 4NT = 6NT \text{ FLOPs.} \tag{8}$$

$$\text{FLOPs}_{\text{step}} = 6 \times N \times T_{\text{step}} \tag{9}$$

By recording the exact number of processed tokens $T$ per step, we compute the cumulative FLOPs reported throughout this paper as the sum of these per-step calculations over the course of training.

### C.2 COEFFICIENT COMPARISON

We consider the two laws

$$\ln L(N, C) = -k_C(N) \ln C + E_C(N), \tag{10}$$

and

$$\ln L(N, D) = -k_D(N) \ln D + E_D(N), \tag{11}$$

are consistent under the linkage $C = ND\phi$ where $\phi > 0$ is a constant for simplification.

**Claim.** Under $C = ND\phi$, the slopes coincide and the intercepts differ by a known shift:

$$k_C(N) = k_D(N) = k(N), \tag{12}$$
$$E_C(N) = E_D(N) + k(N) \ln(N\phi). \tag{13}$$

*Proof.* Substitute $C = ND\phi$ into equation 10:

$$\begin{aligned}
\ln L(N, C) &= -k_C(N) \ln(ND\phi) + E_C(N) \\
&= -k_C(N) \big[\ln D + \ln(N\phi)\big] + E_C(N) \\
&= -k_C(N) \ln D + \Big(E_C(N) - k_C(N) \ln(N\phi)\Big).
\end{aligned}$$

Comparing this with equation 11, i.e., $\ln L(N, D) = -k_D(N) \ln D + E_D(N)$, equality for all $D > 0$ forces the coefficients of $\ln D$ and the constants to match:

$$k_D(N) = k_C(N) =: k(N), \qquad E_D(N) = E_C(N) - k(N) \ln(N\phi).$$

Rearranging the second identity yields equation 13. □

The observation from Figure 4 also matches with this conclusion.

### C.3 FITTING FOR K AND E

Table 5: Uncertainty Analysis for raw data: Base and Instruct Models (Holdout Score)

| Model | Base | | | Instruct | | |
|---|---|---|---|---|---|---|
| | Test Loss | Avg Std | SEM | Test Loss | Avg Std | SEM |
| 0.5B | 0.9419 | 0.0082 | 0.0048 | 0.9458 | 0.0073 | 0.0042 |
| 1B | 0.9129 | 0.0091 | 0.0053 | 0.8988 | 0.0098 | 0.0057 |
| 3B | 0.8582 | 0.0129 | 0.0074 | 0.8281 | 0.0112 | 0.0065 |
| 7B | 0.7148 | 0.0147 | 0.0085 | 0.6777 | 0.0142 | 0.0082 |
| 14B | 0.6051 | 0.0149 | 0.0086 | 0.5588 | 0.0143 | 0.0083 |
| 32B | 0.4937 | 0.0056 | 0.0032 | 0.4579 | 0.0127 | 0.0073 |
| 72B | 0.4359 | 0.0143 | 0.0082 | 0.4320 | 0.0140 | 0.0081 |

Table 6: Comparison of $k_{max}$ and $N_0$ Parameters Across Fitting Scenarios

| Source | Metric | Scenario | $k_{max}$ | $N_0$ (B) | $R^2$ |
|---|---|---|---|---|---|
| Base | L(N,C) | Intra-model | 0.1349 | 13.09 | 0.9955 |
| | L(N,C) | Inter-model | 0.1518 | 17.37 | 0.9944 |
| | L(N,D) | Intra-model | 0.1348 | 11.52 | 0.9953 |
| | L(N,D) | Inter-model | 0.1631 | 16.95 | 0.9947 |
| Instruct | L(N,C) | Intra-model | 0.1276 | 17.27 | 0.9970 |
| | L(N,C) | Inter-model | 0.1443 | 28.33 | 0.9950 |
| | L(N,D) | Intra-model | 0.1325 | 17.08 | 0.9970 |
| | L(N,D) | Inter-model | 0.1484 | 27.15 | 0.9949 |

## D    RESPONSE LENGTH

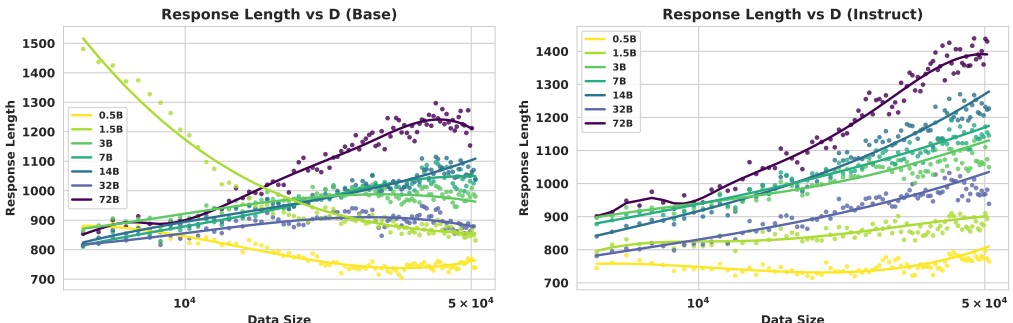

Figure 12: Response length vs. Data size. Left: Base models. Right: Instruct models.

## E    STATEMENTS

### E.1    THE USE OF LARGE LANGUAGE MODELS

We used Large Language Model (LLM) to refine our initial draft. This process included checking for obvious grammatical and syntactical errors, as well as making the language more formal and academic. We reviewed the content generated by the LLM to ensure that no prohibited generated content appeared in the article.

