# OpenReview forum: "Scaling Behaviors of LLM Reinforcement Learning Post-Training: An Empirical Study in Mathematical Reasoning"
_ICLR.cc/2026/Conference — ICLR 2026 Conference Withdrawn Submission_

### Official Review · Reviewer_xzeJ · 2025-10-24

**Soundness:** 3
**Presentation:** 3
**Contribution:** 2
**Rating:** 4
**Confidence:** 3

**Summary:**

The paper conducts an empirical analysis of the effect of different choices in RL fine-tuning for mathematical reasoning. The authors conduct various experiments, varying the model size, compute budget, data size and data repetition. The authors find that larger models train faster and require less data compared to smaller ones, and that data repetition is effective in the data-constrained regime.

**Strengths:**

- The paper focuses on scaling properties of RL for mathematical reasoning, covering some core questions that are of primary interest in the field.
- The authors conduct a rigorous study of different aspects of scaling the RL training, including model size, compute budget and dataset size.
- I find some findings interesting, including the effect of data repetition and number of rollouts.

**Weaknesses:**

- The main issue I see with the paper is that, while the authors run a large set of experiments, the analysis lacks an overall unifying meaningful conclusion. The authors attempt to suggest that the analysis results in some power laws for e.g. compute or data vs. loss, but the data mostly seems very noisy and deviates significantly from the predicted law. In particular, it seems that summarizing the finding as "larger models train faster" captures the more meaningful results of this study, and fitting scaling laws doesn't seem to contribute a lot.
- Another problem in the overall setting is the fact that, since RL is performed on pretrained models (and not "from scratch" like in "standard" scaling law analysis), larger models have better initial accuracy. This means that the larger models will observe more positive rewards from the beginning, and therefore it is expected that they will train faster. I think that an experiment that controls for the initial accuracy, e.g. trains models of different sizes with the same/similar initial accuracy, would be more appropriate for this type of study. Alternatively, the authors could study the same models but on a task where they all have similar initial accuracy (if such a task exists).
- In the introduction of the scaling law analysis, it was not completely clear to me whether the compute considered is only the RL compute, or does it include the pertaining compute as well? I believe that the analysis does not take into account the compute required for pertaining, which makes the overall conclusion unclear.
- Some fitted curves (e.g. Figure 5 and Figure 7) really do not fit the data at all, and it's unclear what they are adding to the analysis.
More minor comments:
- Dataset setting: "We further sort the problems by increasing difficulty (decreasing pass rate)" - according to which model are these sorted?
- Lines 72-74 are hard to read and understand.
- Line 46-47: "scales power-law" => "scales like a power-law"?
- Line 96 - "the key hyperparameter rollout number"
- Line 210 - "making more challenging loss reduction." => "making the loss reduction more challenging"
- I'm not sure why the authors choose to use the term "test loss" instead of "accuracy" or "error", as "loss" is typically used to discuss something like e.g. cross-entropy loss. Using "zero-one loss" would also be better in my opinion.

**Questions:**

See above

---

> ### Author Response · Authors · 2025-11-21
> **Response to Reviewer xzeJ (1/6)**
>
> We are deeply grateful for your insightful and detailed suggestions. We acknowledge your assessment that our original analysis lacked a unifying, meaningful conclusion, and that mere fitting of training points fell short of delivering substantial value. To address these concerns, we have implemented several updates in the revision.
>
> Firstly, **We have expanded our experiments to 32B and 72B models** and, with more comprehensive data, we have derived the analytic formula **that shows a saturation trend for learning efficiency term $k(N)$**, the scaling law now could be revised as:
> $$\log(L) = -k(N) \cdot \log(C) + E(N),\ \text{where} \ \  k(N) = \left(\frac{K_{max}}{1+\frac{N_{0}}{N}}\right)$$
>
> Secondly, the revised scaling law that incorporates the updated $k(N)$ term delivers an excellent fit to the experimental data points across the entire 0.5B–72B parameter range. Beyond mere fitting, our deeper analysis of the empirical law reveals that it also **exhibits a predictive capability** for both inter-model and intra-model aspects, the fiiting results with predition on unseen model, compute and data could be found in Section 3.1 and 3.1.
>
> We will now respond to each of your points in detail.

---

> ### Author Response · Authors · 2025-11-21
> **Response to Reviewer xzeJ (2/6)**
>
> >**W1**：The main issue I see with the paper is that, while the authors run a large set of experiments, the analysis lacks an overall unifying meaningful conclusion. The authors attempt to suggest that the analysis results in some power laws for e.g. compute or data vs. loss, but the data mostly seems very noisy and deviates significantly from the predicted law. In particular, it seems that summarizing the finding as "larger models train faster" captures the more meaningful results of this study, and fitting scaling laws doesn't seem to contribute a lot.
>
> **A1**: To address your concern that the original analysis lacked a unifying, meaningful conclusion, we have leveraged our updated data and deeper analysis to reach stronger, more comprehensive conclusions with actionable utility:
> - **The Efficiency Saturation Effect**: While it is true that larger models exhibit higher learning efficiency, **our analytic $k(N)$ formula refines previous conclusion: efficiency increases with model size, but the marginal gains from expanding parameters diminish gradually**. Rather than growing indefinitely, efficiency follows a predictable saturation curve and converges toward a maximum value $K_{max}$. We put the details in Section 3.1 (Line 224-235), Section 3.2 (Line 279-284) and Figure 4.
> - **The Predictive Capability**: We found that our empirical scaling law with analytic $k(N)$ not only fits the experimental data but also exhibits a predictive capability. As illustrated in Figures 2 and 3 in PDF, the formula provides actionable value by allowing practitioners to:
>     - **Inter-Model Prediction**: Predict large-model training efficiency from smaller, cheaper model runs (e.g., fitting on $\le$ 32B to predict the 72B trend, Figure 2a, 2b, 3a, 3b in Section 3.1 and Section 3.2).
>     - **Intra-Model Prediction**: Predict the training trend from only the initial training data of a single run (e.g., fitting on the first 25% compute to predict the rest of the run, Figure 2c, 2d, 3c, 3d in Section 3.1 and Section 3.2).

---

> ### Author Response · Authors · 2025-11-21
> **Response to Reviewer xzeJ (3/6)**
>
> >**W2**：Another problem in the overall setting is the fact that, since RL is performed on pretrained models (and not "from scratch" like in "standard" scaling law analysis), larger models have better initial accuracy. This means that the larger models will observe more positive rewards from the beginning, and therefore it is expected that they will train faster. I think that an experiment that controls for the initial accuracy, e.g. trains models of different sizes with the same/similar initial accuracy, would be more appropriate for this type of study. Alternatively, the authors could study the same models but on a task where they all have similar initial accuracy (if such a task exists)
>
> **A2**:Our study is designed to investigate the empirical scaling laws within the LLM-based RL Post-Training context. In the real-world scenario, **the differential initial accuracy across models of varying size is an intrinsic and inseparable factor of the model scale, not an experimental confounder**. The primary objective of our research is to find empirical scaling laws that provide practical guidance for standard, public model families (like Qwen2.5). And we have successfully derived such laws with meaningful predictivity, demonstrating that robust patterns can be captured despite differences in initial accuracy.**If we were to artificially control for initial accuracy, the resulting law would ignore this crucial, common factor and limit the practical generalizability**.
>
> Despite the focus of this paper, we acknowledge your proposal to study scaling behavior under a constant initial accuracy is a scientifically valuable and interesting direction. We believe this is an excellent avenue for future research to further characterize the complexities of LLM-based RL scaling.

---

> ### Author Response · Authors · 2025-11-21
> **Response to Reviewer xzeJ (4/6)**
>
> >**W3**:In the introduction of the scaling law analysis, it was not completely clear to me whether the compute considered is only the RL compute, or does it include the pertaining compute as well? I believe that the analysis does not take into account the compute required for pertaining, which makes the overall conclusion unclear
>
> **A3**:You are correct that our analysis only considers the compute used for the RL post-training phase. Our work is designed to investigate scaling behaviors within the LLM-based RL post-training context, with the goal of providing actionable, practical guidance for practitioners working on RL fine-tuning processes. Given this focus, we made an active methodological choice to exclude pre-training compute because **it would conflate the resource allocation needs of pre-training and RL post-training, undermining the law’s ability to deliver targeted, reproducible guidance for the scenario we aim to address**.

---

> ### Author Response · Authors · 2025-11-21
> **Response to Reviewer xzeJ (5/6)**
>
> > **W4**:Some fitted curves (e.g. Figure 5 and Figure 7) really do not fit the data at all, and it's unclear what they are adding to the analysis.
>
> **A4**：These sections serve as **auxiliary components** of our empirical analysis. As a comprehensive empirical study, our goal with Section 3.4 (data reuse) and 3.6 (GRPO rollout size) was to conduct preliminary explorations to observe and quantify how these factors influence the overall training trends and model performance.

---

> ### Author Response · Authors · 2025-11-21
> **Response to Reviewer xzeJ (6/6)**
>
> **Minor Comments**
> > Dataset setting: "We further sort the problems by increasing difficulty (decreasing pass rate)" - according to which model are these sorted?
>
> Thank you for catching this omission. The problems were sorted by the pass rate of the Qwen2.5-7B-Base model. We will add this clarification to the text.
>
> >Lines 72-74 are hard to read and understand. / Line 46-47: "scales power-law" / Line 96 - "the key hyperparameter rollout number" / Line 210 - "making more challenging loss reduction."
>
> Thank you for this detailed proofreading. We have corrected these writing problems and other typographical errors in the revised manuscript.
>
> > I'm not sure why the authors choose to use the term "test loss" instead of "accuracy" or "error" This is a fair point on terminology.
>
> As defined in the paper (Section 2, Line 160), our Test Loss is a 0-1 loss (specifically, 1 - pass_rate). We chose this term just to maintain direct consistency with foundational neural scaling law papers[1], which are also framed in terms of "Loss" vs. "Compute." We have already add a sentence to reinforce this definition to **avoid confusion from cross entropy loss** at Line 164.
>
> Thanks for your reading.
>
> **Reference**
>
> [1] Kaplan, Jared et al. “Scaling Laws for Neural Language Models.” ArXiv abs/2001.08361 (2020): n. pag.

---

> ### Author Response · Authors · 2025-11-27
> **Looking forward to your reply**
>
> Dear Reviewer xzeJ,
>
> **As the discussion window closes in less than a week**, we kindly ask for a moment of your time to review our rebuttal.
>
> We have added 72B experiments and derived the analytical form of $k(N)$. Based on this, we identified the saturation effect of learning efficiency and validated the predictive capability of our scaling law.
>
> We hope these new results warrant a re-evaluation of our work.
>
> Best regards, The Authors

---

### Official Review · Reviewer_Wme4 · 2025-11-01

**Soundness:** 2
**Presentation:** 2
**Contribution:** 3
**Rating:** 2
**Confidence:** 4

**Summary:**

This paper studies scaling RL post training in models under various regimes (compute constrained, data constrained) for mathematical reasoning. The authors do post training using Qwen 2.5 across all possible sizes offered on the mathematics subset of the guru-RL-92k and evaluate on several benchmarks.

**Strengths:**

The main strengths of this paper are:
- the authors run the evals at several model sizes (from 500M to 14B), which can be considered reasonably large for an academic setting
- the authors ablate over the number of rollouts in GRPO which has been shown in literature to be very important for performance
- if I understand correctly, in line 169 the authors say they repeated each config 3 times across seeds - this is very much appreciated since in RL in general the seed can cause quite large differences

**Weaknesses:**

The main weaknesses:
- as far as I understand, the authors did not sweep over hyperparameters. It is not clear to me why these exact values for the learning rate etc were used
- the metrics reported are insufficient. From my understanding, the authors only report pass@1. For a more comprehensive overview, it would have been better to see pass@64, and majority@1 and majority@64
- I am not sure why the plots do not have standard deviations plotted as well as it seems to be possible from the collected data

As a separate point, my main concern is on section 3.4. Unless I am misunderstanding the figures, 100 steps of training is too few training steps to draw any conclusion about the behaviour. I am curious if these plots would still hold at much longer training times and if the trends would swap. It seems from figure 4 that you do have enough data available to train for much longer.

**Questions:**

- In section 3.4, for tau=100, given that you train for 100 steps, does this mean you repeat a single batch 100 times? If so, for how many rollouts?
- In section 3.3, what does it mean that you train (referring to post-training in this setting) to convergence? Are you referring to doing GRPO on the guru-RL-92k dataset until the average reward hits 1?

---

> ### Author Response · Authors · 2025-11-21
> **Response to Reviewer Wme4 (1/6)**
>
> Dear Reviewer Wme4:
>
> Thank you for your detailed review. Your comments are very important for us to revise our work.
>
> Our research aims to investigate the empirical scaling behavior of mathematical reasoning in LLM post-training, through an empirical study conducted across the **0.5B–72B** model. **We found that efficiency gains saturate as model size increases, and the law we derived exhibits predictive capability through both inter-model extrapolation and intra-model forecasting**.
>
> We will now address your specific critiques in detail.

---

> ### Author Response · Authors · 2025-11-21
> **Response to Reviewer Wme4 (2/6)**
>
> > **W1**:as far as I understand, the authors did not sweep over hyperparameters. It is not clear to me why these exact values for the learning rate etc were used.
>
> **A1**: When conducting our experiments, we prioritized stability and simplicity by directly **adopting the nearly same hyperparameter settings used by the Guru dataset authors** for their RL experiments (https://github.com/LLM360/Reasoning360/tree/main/scripts/train), and we kept these hyperparameters consistent across all model sizes (0.5B to 72B) to eliminate them as a confounding variable and isolate the true effect of model scale.

---

> ### Author Response · Authors · 2025-11-21
> **Response to Reviewer Wme4 (3/6)**
>
> >**W2**:the metrics reported are insufficient. From my understanding, the authors only report pass@1. For a more comprehensive overview, it would have been better to see pass@64, and majority@1 and majority@64
>
> **A2**: We acknowledge that the those metrics are widely used in RL papers. However, to effectively investigate the scaling behavior, $\text{Test Loss} = 1 - \text{Pass}@1$ still serves as the most direct and fundamental signal of the model's intrinsic capability. Using metrics like $\text{pass}@k$ or $\text{majority}@k$ would introduce sampling-based variance and an additional hyperparameter ($k$), thereby undermining our effort to find a smooth, clean empirical scaling law.

---

> ### Author Response · Authors · 2025-11-21
> **Response to Reviewer Wme4 (4/6)**
>
> >**W3**:I am not sure why the plots do not have standard deviations plotted as well as it seems to be possible from the collected data
>
> **A3**: To analyze the data more comprehensively, we have incorporated the standard deviation as you suggested, along with other statistical information such as the Standard Error of the Mean (SEM). They are all listed in the table below, and **could be found in Appendix C.3**:
>
> Table 1: Uncertainty Analysis: Base - holdout_score
> | Model | Test Loss | Avg Std | SEM  |
> | :--- | :---: | :---: | :---: |
> | 0.5B | 0.9419 | 0.0082 | 0.0048 |
> | 1.5B | 0.9129 | 0.0091 | 0.0053 |
> | 3B | 0.8582 | 0.0129 | 0.0074 |
> | 7B | 0.7148 | 0.0147 | 0.0085 |
> | 14B | 0.6051 | 0.0149 | 0.0086 |
> | 32B | 0.4937 | 0.0056 | 0.0032 |
> | 72B | 0.4359 | 0.0143 | 0.0082 |
>
> Table 2: Uncertainty Analysis: Instruct - holdout_score
> | Model | Final ErrRate | Avg Std | SEM |
> | :--- | :---: | :---: | :---: |
> | 0.5B | 0.9458 | 0.0073 | 0.0042 |
> | 1.5B | 0.8988 | 0.0098 | 0.0057 |
> | 3B | 0.8281 | 0.0112 | 0.0065 |
> | 7B | 0.6777 | 0.0142 | 0.0082 |
> | 14B | 0.5588 | 0.0143 | 0.0083 |
> | 32B | 0.4579 | 0.0127 | 0.0073 |
> | 72B | 0.4320 | 0.0140 | 0.0081 |

---

> ### Author Response · Authors · 2025-11-21
> **General Response to Reviewer Wme4 (5/6)**
>
> > **W4**:As a separate point, my main concern is on section 3.4. Unless I am misunderstanding the figures, 100 steps of training is too few training steps to draw any conclusion about the behaviour. I am curious if these plots would still hold at much longer training times and if the trends would swap.
>
>
> **A4**:The experimental design in this section holds the **total number of training data, and batch size fixed across all runs**. For example, a run with a reuse factor of $\tau=2$ would sample $1/2$ of the unique data (while preserving the difficulty curriculum) and repeat this subset twice, such that $D_{unique} \times \tau = D_{const}$.
>
> Additionally, this section is intended as an **auxiliary exploration**, forming an part of our comprehensive empirical investigation. Its primary goal was to conduct a qualitative assessment of the impacts arising from data reuse, with its takeaway being a preliminary finding that moderate reuse does not appear to undermine generalization. Given the inherent constraints on our computing capacity, we did not allocate substantial computational resources to train them for longer steps.

---

> ### Author Response · Authors · 2025-11-21
> **Response to Reviewer Wme4 (6/6)**
>
> >**Q1**:In section 3.4, for tau=100, given that you train for 100 steps, does this mean you repeat a single batch 100 times? If so, for how many rollouts?
>
> **A1**:A step in this plot refers to one full rollout-and-update cycle. As detailed in our response to Reviewer 5U2P, one cycle involves rolling out 512 prompts(Batch Size $= 512$) with a group size of 8 ($512 \times 8 = 4,096$ responses). Therefore, $\tau=100$ for 100 steps means: $100 \text{ steps} \times (512 \text{ prompts/step} \times 8 \text{ responses/prompt}) = 409,600$ total responses were processed, where the unique data subset was repeated 100 times.
>
> >**Q2**:In section 3.3, what does it mean that you train (referring to post-training in this setting) to convergence? Are you referring to doing GRPO on the guru-RL-92k dataset until the average reward hits 1?
>
> **A2**:"Convergence" here does not mean the average reward hits 1 (or 0% error rate). In our main experiments, it means we trained the models until the test loss on the held-out validation set visibly plateaued and no longer showed significant rapid improvement. At that point, we stopped training to allocate compute resources to the other experiments in our study.
>
> Thanks for your reading.

---

> ### Author Response · Authors · 2025-11-27
> **Looking forward to your reply**
>
> Dear Reviewer Wme4,
>
> Thank you for your detailed review.
>
> **As the rebuttal deadline approaches in less than a week**, we are eager to hear your thoughts on our response. We have provided clarifications on our hyperparameter choices  and have added comprehensive experiments using the 32B and 72B model to strengthen our empirical evidence.
>
> We kindly ask you to reconsider our work in light of these updates. We are standing by to answer any final questions you may have.
>
> Best regards, The Authors

---

### Official Review · Reviewer_5U2P · 2025-11-03

**Soundness:** 2
**Presentation:** 3
**Contribution:** 2
**Rating:** 4
**Confidence:** 4

**Summary:**

This work addresses establishing scaling laws for reinforcement learning (RL) post-training on LLMs for mathematical reasoning tasks. Across 54 Qwen2.5 experiments (0.5B–14B), the authors find that larger models are both more compute- and data-efficient, and that performance depends mainly on total optimization steps rather than data uniqueness, making data reuse highly effective. RL improves in-domain math reasoning but transfers poorly to other domains, and scaling behaviors remain consistent across base and instruct models. Overall, the work highlights the following: favor larger models, reuse high-quality data, and tune rollout size to compute budget.

**Strengths:**

- Establishing general and actionable scaling laws for the RL post-training setting for LLMs is an important problem.
- The authors repeat three runs for each configuration in their results.
- The specificity of the data reuse results, and the observation about the gap between dense models and mixture of experts models are interesting.

**Weaknesses:**

- The work’s findings are confined to a single model family (Qwen2.5) and math dataset with an assigned curriculum. The extent of the generality of the specificity of their findings (eg. the fitted scaling law) since RL post-training behavior is highly sensitive to reward design, data composition, and curriculum—unlike the more standardized and robust scaling behavior seen in pretraining.
- Furthermore, it’s unclear what the actionable takeaways from this work are; like the authors write in the introduction, “these laws … are invaluable... because they provide actionable guidance on how to distribute scarce computational resources most effectively”. The observations made in each section are too broad to be actionable (perhaps with the exception of Observation 4) and already reflect the general sentiment that if one has the computational resources, starting RL fine-tuning from a larger model is more beneficial.
- While this paper identifies empirical scaling trends for RL post-training, I believe the real practical value of scaling laws for RL lies in whether they’re able to provide predictive insights. To be more specific, whether one can forecast when additional rollout compute will continue to yield gains or when performance will plateau. This work fits power-law curves post-hoc but does not test whether these fits can extrapolate to unseen compute budgets or reliably guide future training runs.
- I believe the work would benefit from analyzing additional metrics like pass@k for their runs; it is also of interest to see whether model capability is improving under this metric (eg. are larger models reliably able to solve more problems that their base model previously couldn't have even under multiple samples)?

Minor:
- Line 313: toal -> total

**Questions:**

- Is the normalized error rate used for the scaling law calculated with respect to greedy sampling?
- How does group size and the mini batch size used for the GRPO actor update relate to the batch size of 512? Specifically how many offline update steps are being taken with respect to the rollout policy?
- What do the pass@k (or equivalent reward-based) scaling curves look like under your setup? Do they follow similar power-law behavior as the test loss curves you report?
- Could the authors elaborate on what they see as the main conceptual or practical value of these results? In particular, how should practitioners interpret their findings when deciding how to allocate RL compute or data?

---

> ### Author Response · Authors · 2025-11-21
> **Response to Reviewer 5U2P (1/5)**
>
> Dear Reviewer 5U2P:
>
> Thank you for reading our paper and providing valuable guidance. We fully understand your concerns regarding the generality of our scaling law, its practical takeaways, and most importantly its predictive power. In response to your feedback, **we have conducted additional experiments with 32B and 72B models to enhance generality, performed a deeper analysis of the learning efficiency term $k(N)$, and validated the scaling law’s predictive power**, specifically addressing these concerns.

---

> ### Author Response · Authors · 2025-11-21
> **Response to Reviewer 5U2P (2/5)**
>
> > **W1**:The work’s findings are confined to a single model family (Qwen2.5) and math dataset... RL post-training behavior is highly sensitive to reward design, data composition, and curriculum...
>
> **A1**:As you noted, RL post-training behavior is task-dependent, sensitive to reward design and many other factors. To derive a clear empirical scaling law, our paper was designed to isolate the effect of parameter count by eliminating confounding variables, opted for a single model family to avoid architectural biases, and centered on mathematics with a consistent rule-based reward to mitigate noise and task-interference effects.

---

> ### Author Response · Authors · 2025-11-21
> **Response to Reviewer 5U2P (3/5)**
>
> > **W2&3**:...it’s unclear what the actionable takeaways from this work are... I believe the real practical value of scaling laws for RL lies in whether they’re able to provide predictive insights... This work fits power-law curves post-hoc but does not test whether these fits can extrapolate..
>
> **A2&3**:To address the need for actionable takeaways and predictive insights, we expanded our experiments to include 32B and 72B models and derived an analytic formula for learning efficiency. Now the previous formula is has been revised as:
> $$\log(L) = -k(N) \cdot \log(C) + E(N),\ \text{where} \ \  k(N) = \left(\frac{K_{max}}{1+\frac{N_{0}}{N}}\right)$$
> Based on this, our actionable takeaways are:
> - **Efficiency Saturates**: While larger models are more efficient, this gain diminishes and eventually saturates to $K_{max}$. As shown in Table 1 below, the efficiency gain from 0.5B to 14B is massive ($+0.062$), whereas the gain from 32B to 72B is far less pronounced ($+0.019$). And we put the details of the saturation effect at Section 3.1 (Line 224-235), Section 3.2 (Line 279-284) and Figure 4.
>
> Table 1: K(N) for Base Models
> | Model Size | $k_C(N)$ (Compute) | $k_D(N)$ (Data) |
> | :--- | :---: | :---: |
> | 0.5B | 0.00466 | 0.00518 |
> | 1.5B | 0.01306 | 0.01448 |
> | 3B | 0.02378 | 0.02624 |
> | 7B | 0.04481 | 0.04898 |
> | 14B | 0.06703 | 0.07255 |
> | 32B | 0.09296 | 0.09948 |
> | 72B | 0.11162 | 0.11849 |
>
> - **The Law is Predictive**: We validated that the revised law can guide training through:
>     - Inter-Model Prediction: Fitting on models $\le$ 32B successfully extrapolates the training efficiency $k$ for the 72B model.(Figure 2a, 2b, 3a, 3b in Section 3.1 and Section 3.2)
>     - Intra-Model Prediction: Fitting on initial training data can predicts the model's future trajectory.(Figure 2c, 2d, 3c, 3d in Section 3.1 and Section 3.2)
> - Other Practical Guidance on Resource Allocation (Section 3.1 Line 231-235):
>     - Small-Scale Regime: "Bigger is better" is dominant due to massive efficiency gains.
>     - Large-Scale Regime: Due to the saturation effect, practitioners with fixed compute budgets should consider balancing model size and training steps. For instance, in mathematical reasoning, since the 32B and 72B efficiencies are similar (0.09 vs 0.11), it may be better to train a 32B model for more steps rather than a 72B model for fewer steps under limited compute budget.

---

> ### Author Response · Authors · 2025-11-21
> **Response to Review 5U2P (4/5)**
>
> > **W4&Q3**:I believe the work would benefit from analyzing additional metrics like pass@k... What do the pass@k... scaling curves look like...
>
> **A4**: We acknowledge that $\text{pass}@k$ is a widely used metric in evaluating the model's full capability, but our goal was not to report a new SOTA performance. We aimed to understand the scaling properties in mathematical reasoning. In this case, we believe $\text{Test Loss} = 1 - \text{Pass}@1$ serves as the most direct and fundamental metric. Additionaly, similar works like OpenAI’s technical report ["learning to reason with llms"](https://openai.com/index/learning-to-reason-with-llms/) also use Pass@1 as their metric.

---

> ### Author Response · Authors · 2025-11-21
> **Response to Reviewer 5U2P (5/5)**
>
> > **Q1**：Is the normalized error rate used for the scaling law calculated with respect to greedy sampling?
>
> No. Validation uses sampling with temperature=0.7 and do sample=True, not greedy decoding.
>
> > **Q2**:How does group size and the mini batch size used for the GRPO actor update relate to the batch size of 512? Specifically how many offline update steps are being taken with respect to the rollout policy?
>
> Rollout collects 512 prompts × 8 responses = 4,096 responses, split into mini-batches of 128 responses each, resulting in 32 mini-batches and **32 offline gradient steps**.
>
> **Q3** have been answered with **W4&Q3** above.
>
> > **Q4**:Could the authors elaborate on what they see as the main conceptual or practical value of these results? In particular, how should practitioners interpret their findings when deciding how to allocate RL compute or data?
>
> The main value of our work is encapsulated in the **predictive scaling law derived from our empirical analysis**. The value can be summarized in two key points:
> 1. **The Saturation Effect**: The analytic formula for $k(N)$ provides an insight into how RL post-training scales: while larger models consistently exhibit higher learning efficiency, this efficiency gain is not infinite. The formula $k(N) = \left(\frac{K_{max}}{1+\frac{N_{0}}{N}}\right)$ quantitatively shows that the efficiency gains from increasing parameter size diminish and eventually saturate towards a maximum limit $K_{max}$.
>
> 2. **The Predictive Power**: The law is not merely a post-hoc fit but a predictive tool. As detailed in our paper Sections 3.1 and 3.2, the formula delivers actionable value: it enables practitioners to predict the performance of large models from smaller, more cost-effective model runs (inter-model extrapolation), or to forecast the full training trajectory using only the initial training data of a single run (intra-model prediction).
>
> Thanks for your reading.

---

> ### Author Response · Authors · 2025-11-27
> **Looking forward to your reply**
>
> Dear Reviewer 5U2P,
>
> Thank you for your helpful comments.
>
> **With less than a week remaining in the discussion period**, we kindly hope you can take a moment to review our updates. To address your concerns about generality, we have extended our analysis to include 32B and 72B models, covering the full Qwen2.5 series, and demonstrated the predictive capability to provide actionable takeaways.
>
> Your feedback on these new results would be incredibly valuable to us before the window closes.
>
> Best regards, The Authors

---

### Official Review · Reviewer_MN3w · 2025-11-06

**Soundness:** 3
**Presentation:** 3
**Contribution:** 2
**Rating:** 4
**Confidence:** 4

**Summary:**

This paper presents an empirical study of reinforcement-learning (RL) post-training for mathematical reasoning across Qwen2.5 models from 0.5B to 14B parameters. Using a consistent GRPO setup and ~50k curated math problems, the authors probe three regimes—compute-constrained, data-constrained, and data-reuse—over 50+ controlled runs. The central findings are: (i) under fixed compute, larger models trained for fewer steps achieve lower test loss and steeper loss–compute slopes; (ii) with fixed unique data, larger models are more sample-efficient and follow a power-law loss–data relation; (iii) when trained to convergence on sufficiently large data, performance improves monotonically with model size, though small models deviate from the idealized fits; and (iv) with steps held fixed, moderate data repetition has little impact, while extreme reuse leads to overfitting. An ablation on GRPO rollout group size shows larger groups improve sample efficiency, but the compute-optimal choice depends on budget. The paper’s stated contribution is practical guidance for allocating compute, data, and reuse when performing RL post-training on reasoning tasks. While the execution is careful and the results are useful for practitioners, the study primarily re-validates familiar scaling intuitions (bigger models, more data, cautious repetition) and offers limited technique-level insight specific to RL post-training.

**Strengths:**

1. All experiments use one model family (Qwen2.5) and one RL algorithm (GRPO) with matched tokenization, prompting, and reward extraction. This reduces design variance and makes cross-run comparisons credible, letting readers attribute differences to scale/data rather than to confounders (architecture swaps, reward model changes, or prompt drift).

2. The compute-constrained, data-constrained, and reuse-constrained formulations are clearly defined and answered with targeted sweeps. Practitioners get concrete guidance: prefer bigger models at fixed FLOPs, expect diminishing returns from unique data vs. moderate repetition, and tune rollout group size to budget. These are decisions teams regularly face in RL post-training.

3. The paper does not stop at in-distribution accuracy; it measures transfer to code/logic/STEM, reports degradation cases, and includes an efficiency ablation (rollout size). This candor about failure modes and trade-offs increases the trustworthiness and operational value of the study.

**Weaknesses:**

1. The main results mirror known pretraining/data-centric scaling laws: larger models win under fixed compute; power-law data scaling; moderate repetition is fine until it isn’t. And the
The paper offers little RL-specific theory (e.g., KL/reward-landscape analysis, credit-assignment diagnostics) to explain why these slopes arise, so the contribution feels incremental.

2. Although the paper tries to figure the scaling laws for reinforcement learning, there lacks of experiments to demonstrate the effectiveness in "predictable scaling" of the proposed laws and whether it can help the scaling trends of RL training.

3. All results come from a single dense family ≤14B on math-centric data. Without diversity in model families (MoE, larger dense), domains, or data curricula, it’s unclear whether the conclusions persist beyond the studied slice.

4. The core “test loss” is derived from Pass@1 extraction on a small (≈500) in-distribution holdout and may be sensitive to parsing/prompt choices. Confidence intervals and variance across random seeds are sparse; compute/data accounting under heavy reuse isn’t fully audited.

**Questions:**

Please refer to the weakness

---

> ### Author Response · Authors · 2025-11-21
> **Response to Reviewer MN3w (1/5)**
>
> Dear Reviewer MN3w:
>
> We would like to express our sincere gratitude for your highly valuable suggestions for our paper. We have **expanded the scope of our experiments to 72B** and built upon this more comprehensive data to conduct a deeper analysis of $k(N)$ and our entire scaling law. Furthermore, we have now supplemented our analysis with more statistical metrics. Our detailed responses are as follows:

---

> ### Author Response · Authors · 2025-11-21
> **Response to Reviewer MN3w(2/5)**
>
> > **W1**:The main results mirror known pretraining/data-centric scaling laws: larger models win under fixed compute; power-law data scaling; moderate repetition is fine until it isn’t. And the The paper offers little RL-specific theory (e.g., KL/reward-landscape analysis, credit-assignment diagnostics) to explain why these slopes arise, so the contribution feels incremental
>
> **A1**: With more comprehensive data, we have moved a step forward by deriving an analytic formula for the $k(N)$ term:
> $$k(N) = \left(\frac{K_{max}}{1+\frac{N_{0}}{N}}\right)$$
> It demonstrates a conclusion that though larger models consistently exhibit higher learning efficiency, the efficiency (slope $k$) will not increase indefinitely with model size but saturates towards a maximum limit $K_{max}$. The details can be found in Section 3.1, Section 3.2 and Figure 4.
>
> Regarding the theoretical underpinnings, we acknowledge that we have not yet found a definitive explanation for why the scaling follows this form. We hope that empirically establishing and quantifying this saturation effect provides a useful foundation for future theoretical research

---

> ### Author Response · Authors · 2025-11-21
> **Response to Reviewer MN3w(3/5)**
>
> > **W2**:Although the paper tries to figure the scaling laws for reinforcement learning, there lacks of experiments to demonstrate the effectiveness in "predictable scaling" of the proposed laws and whether it can help the scaling trends of RL training.
>
> **A2**:Using the analytic $k(N)$ formula, the scaling law could be revised as:
> $$\log(L) = -k(N) \cdot \log(C) + E(N),\ \text{where} \ \  k(N) = \left(\frac{K_{max}}{1+\frac{N_{0}}{N}}\right)$$
> we validated this capability in two key ways:
> - **Inter-Model Extrapolation** (Predicting Larger Models):We fitted the scaling law parameters using only data from 0.5B-32B models and used this to successfully predict the training slope of the unseen 72B model. (as shown in figure 2a/2b, 3a/3b in Section 3.1 and Section 3.2)
> - **Intra-Model Prediction** (Predicting Future Training): We fitted the law using only the initial training data and found it accurately predicted the model's performance for the remainder of the run. (as shown in figure 2c/2d, 3c/3d in Section 3.1 and Section 3.2)

---

> ### Author Response · Authors · 2025-11-21
> **Response to Reviewer MN3w (4/5)**
>
> > **W3**:All results come from a single dense family ≤14B on math-centric data. Without diversity in model families (MoE, larger dense), domains, or data curricula, it’s unclear whether the conclusions persist beyond the studied slice.
>
> **A3**:We have now addressed this limitation by expanding our study to include 32B and 72B models, covering the full Qwen2.5 dense model series (0.5B to 72B, **detailed in Figure 1 and the introduction Line 72**). We confirm our findings, including the saturation effect, hold across a much wider $\sim$ 140x range of parameters.
>
> Regarding the use of a single model family and domain, we think that introducing different model families would also introduce confounding variables from structural and architectural differences, creating a bias we sought to avoid. Similarly, since RL performance can be task-dependent, we focused on a single domain to establish a controlled scaling standard without the interference  of other domains.

---

> ### Author Response · Authors · 2025-11-21
> **Response to Reviewer MN3w (5/5)**
>
> >**W4** The core “test loss” is derived from Pass@1 extraction on a small (≈500) in-distribution holdout and may be sensitive to parsing/prompt choices. Confidence intervals and variance across random seeds are sparse; compute/data accounting under heavy reuse isn’t fully audited.
>
> **A4**:Thank you for pointing out the need for more rigorous statistical analysis. We note that calculating 95\% confidence intervals (CIs) based solely on three random seeds would be insufficiently reliable (three samples per step are too limited to capture the full variability), so we have supplemented statistical analyses for additional metrics. These results, including relevant variance assessments, **are detailed in Appendix C.3 (Line 1192)** and listed as follows:
>
> Table 1: Uncertainty Analysis: Base - holdout_score
> | Model | Final Test Loss  | Avg Std  | SEM  |
> | :--- | :---: | :---: | :---: |
> | 0.5B | 0.9419 | 0.0082 | 0.0048 |
> | 1.5B | 0.9129 | 0.0091 | 0.0053 |
> | 3B | 0.8582 | 0.0129 | 0.0074 |
> | 7B | 0.7148 | 0.0147 | 0.0085 |
> | 14B | 0.6051 | 0.0149 | 0.0086 |
> | 32B | 0.4937 | 0.0056 | 0.0032 |
> | 72B | 0.4359 | 0.0143 | 0.0082 |
>
> Table 2: Uncertainty Analysis: Instruct - holdout_score
> | Model | Final Test Loss | Avg Std  | SEM  |
> | :--- | :---: | :---: | :---: |
> | 0.5B | 0.9458 | 0.0073 | 0.0042 |
> | 1.5B | 0.8988 | 0.0098 | 0.0057 |
> | 3B | 0.8281 | 0.0112 | 0.0065 |
> | 7B | 0.6777 | 0.0142 | 0.0082 |
> | 14B | 0.5588 | 0.0143 | 0.0083 |
> | 32B | 0.4579 | 0.0127 | 0.0073 |
> | 72B | 0.4320 | 0.0140 | 0.0081 |
>
> Thanks for your reading.

---

> ### Author Response · Authors · 2025-11-27
> **Looking forward to have further discussion with our updates**
>
> Dear Reviewer MN3w,
>
> Thank you again for your time.
>
> **As the discussion period concludes in less than a week**, we kindly hope you can take a moment to review our updates. We have specifically addressed your concerns by expanding our experiments to the 72B model, deriving a formula for learning efficiency, and validating the predictive power of our Scaling Law.
>
> We hope these updates resolve your concerns and we look forward to hearing from you before the deadline.
>
> Best regards, The Authors

---

### Author Response · Authors · 2025-11-21
**General Response**

Dear AC and Reviewers,

We sincerely thank you for your detailed and constructive comments and feedback on our manuscript. We have carefully reviewed your valuable suggestions and identified the primary concerns, which centered on the generality of our experimental scope, **the predictive power of the scaling laws, the clarity of our main conclusions**, and the need for a more rigorous statistical analysis.

To directly address these points and strengthen our findings, we have performed an revision of our work, incorporating new experiments and deeper analysis:
- **Expanded Experimental Scope to 72B**: We expanded our experiments to include the **32B** and **72B** models (both Base and Instruct models). Our study now covers the entire Qwen2.5 dense model series.
- **Analytic Form for Learning Efficiency $k(N)$ Revealing Saturation**: We proposed an analytic form for the learning efficiency coefficient $k(N)$ base on deeper analysis of comprehensive data. It demonstrates that the learning efficiency gains from model scale follow a saturation curve towards a maximum limit $K_{\text{max}}$, as shown in **Section 3.1, Section 3.2** and **Figure 4**.
- **Validated the predictive power of the empirical scaling law**:
    - Inter-Model Extrapolation: Successfully predicting the training efficiency of the 72B model by fitting on 0.5B-32B model data (as shown in the new Figures 2a/2b and 3a/3b in **Section 3.1 and 3.2**)
    - Intra-Model Forecasting: Accurately projecting a single model's future training trajectory based on its initial performance data (as shown in the new Figures 2c/2d and 3c/3d in **Section 3.1 and 3.2**)
- **Enhanced Statistical Analysis**: We added new statistical analysis for the test loss and fitted parameters, which is now detailed in the **Appendix C.3**.

We also utilized new data to update and refine the discussion and figures in Section 3.3 and 3.5, as well as other accompanying appendices/figures/tables.

**All updates are colored in red in the revised PDF**.

These updates allow us to **strengthen our previous contribution by following**:
- **Efficiency Saturates**: With given analytic K(N) formula, we can see that larger models do consistently exhibit higher learning efficiency, but the gain brought by model parameter follows a predictable saturation curve to max learning speed $K_{max}$.
- **The Law is Predictive**: The log-log scale formula describing scaling behaviors is predictive, allowing for reliable extrapolation.

Thanks for reading our rebuttal and looking forward to your reply.

---

### Note · Authors · 2025-12-18

I have read and agree with the venue's withdrawal policy on behalf of myself and my co-authors.